# Systematic functional interrogation of SARS-CoV-2 host factors using Perturb-seq

Sara Sunshine[1], Andreas S. Puschnik [2], Joseph M. Replogle [3,4],
Matthew T. Laurie[1], Jamin Liu [1,5], Beth Shoshana Zha [6], James K. Nuñez[3,7],
Janie R. Byrum[2], Aidan H. McMorrow[2], Matthew B. Frieman [8],
Juliane Winkler [9,10], Xiaojie Qiu[4,11], Oren S. Rosenberg [12],
Manuel D. Leonetti [2], Chun Jimmie Ye [13,14,15,16,17],
Jonathan S. Weissman [4,11,18,19] ✉, Joseph L. DeRisi[1,2] ✉ &
Marco Y. Hein [2,3,20,21,22] ✉

Genomic and proteomic screens have identified numerous host factors of SARS-CoV-2, but efficient delineation of their molecular roles during infection remains a challenge. Here we use Perturb-seq, combining genetic perturbations with a single-cell readout, to investigate how inactivation of host factors changes the course of SARS-CoV-2 infection and the host response in human lung epithelial cells. Our high-dimensional data resolve complex phenotypes such as shifts in the stages of infection and modulations of the interferon response. However, only a small percentage of host factors showed such phenotypes upon perturbation. We further identified the NF-κB inhibitor IκBα (NFKBIA), as well as the translation factors EIF4E2 and EIF4H as strong host dependency factors acting early in infection. Overall, our study provides massively parallel functional characterization of host factors of SARS-CoV-2 and quantitatively defines their roles both in virus-infected and bystander cells.

The coronavirus disease 2019 (COVID-19) pandemic, caused by SARS-CoV-2, has claimed millions of lives and remains a global health burden. Despite the success of rapid vaccine developments, barriers including vaccine access and uptake, as well as breakthrough infections make it imperative to develop both effective antivirals, and therapies targeting an overactive host immune response. A detailed understanding of the host determinants of infection, and the host response throughout infection will broadly inform efforts to develop novel antiviral agents.

Many studies have identified candidate host factors by an array of high-throughput methods, including protein–protein and protein-RNA interaction mapping, as well as CRISPR-based genetic screening[1–10]. Additionally, the host response to SARS-CoV-2 infection has been investigated in single-cell transcriptional studies of blood, bronchial lavage, and tracheal aspirate from COVID-19 patients, human and animal (non-human primate, hamster, ferret) models of infection,

and in cell lines infected in tissue culture[11–18]. However, it remains a challenge to validate individual candidate host factors, delineate their specific roles during infection, and evaluate their suitability as targets for interventions.

Here, we use Perturb-seq, an approach that characterizes the outcomes of CRISPR-based genetic perturbations by single-cell transcriptomics[19–22], to understand how perturbations of host factors alter the course of SARS-CoV-2 infection and the host transcriptional response in human lung epithelial cells. First we compiled a compendium of 183 host factors that were previously reported to either physically interact with viral proteins, score as protective hits in coronaviral genetic screens, or factors that are known for their roles in antiviral host defense pathways[1,2,4–9,23–29]. While all those factors had prior experimental evidence assigning them a role in coronavirus biology, only a small subset of them had been functionally validated. To test how inactivation of each of these genes alters SARS-CoV-2

---

infection dynamics in a highly multiplexed fashion, we subsequently designed a Perturb-seq library to target each of these factors. We performed a CRISPR interference (CRISPRi)[30] experiment in human lung carcinoma (Calu-3) cells, infected with a clinical isolate of SARS-CoV-2 collected in late-2020 (PANGO lineage B.1.503), and subsequently subjected the cells to single-cell RNA sequencing, capturing both infected and uninfected bystander cells. Our results identify transcriptionally distinct clusters of infected and bystander cells, uncover new roles of genetic perturbations in interferon signaling, and functionally validate specific SARS-CoV-2 host dependency factors.

## Results

### Functional genomics of coronavirus host factors with a single-cell readout

To characterize the single-cell transcriptional response to SARS-CoV-2 infection and simultaneously test the effect of host genetic perturbations on viral RNA production and host response, we used Perturb-seq[19–22]. Perturb-seq combines CRISPR-based genetic perturbations with a rich, single-cell transcriptomics readout that is capable of capturing high-dimensional phenotypes, making it well-suited for studying virus-host systems[31]. Viral infection leads to a heterogeneous response in a cell population, characterized, for instance, by cells being in different stages of infection and showing varying levels of activity of antiviral pathways[17,32,33]. Targeting critical host factors can cause shifts in the distribution of cellular states, which delivers insight into the function of any given host factor.

We performed our experiments in Calu-3 cells, a human respiratory epithelial cell line that endogenously expresses the entry receptor of SARS-CoV-2, *ACE2*, albeit at low levels, and has been previously used for several CRISPR screening and single-cell studies

of SARS-CoV-2[17,27,28]. We employed Calu-3 cells engineered to stably express the machinery for CRISPR interference (Methods)[30,34,35]. CRISPRi is highly efficient at suppressing gene expression of selected targets without introducing double-strand breaks, with minimal off-target effects. On-target activity can be maximized by using two single guide RNAs (sgRNAs) per target, expressed from one lentiviral vector[36–38].

We compiled a list of host factors from the literature on SARS-CoV-2 and other coronaviruses, mainly genes identified as protective hits in genetic screens for modifiers of SARS-CoV-2 or related coronavirus infections, and host proteins that were found to interact with viral proteins. We prioritized candidates with multiple lines of evidence supporting their roles in coronavirus biology. Additionally, we curated a list of factors involved in the innate immune response. Overall, we designed and cloned a library containing 239 elements, of which 195 target a single gene, 22 target combinations of two genes (typically paralogs or members of the same pathway, e.g., *ACE2* + *TMPRSS2* or *IFNAR1* + *IFNAR2*), and 22 non-targeting controls (Supplementary Data 1). We packaged the library into lentivirus and delivered it into the engineered Calu-3 cells at a low multiplicity of infection, followed by selection for cells with successful lentivirus integration.

We infected the resulting population for 24 h with a late-2020 clinical isolate of SARS-CoV-2 featuring only a single spike mutation, D614G, and 10 non-synonymous mutations in other genes relative to the ancestral isolate (PANGO lineage B.1.503, complete genome available at GISAID accession ID: EPI_ISL_13689582). Single-cell transcriptomes were then captured using a droplet-based microfluidic workflow (10x Genomics) with direct capture of sgRNAs to reveal which gene or gene pair was targeted in each cell[36] (Fig. 1A). After quality control filtering (Methods), we profiled the transcriptomes of

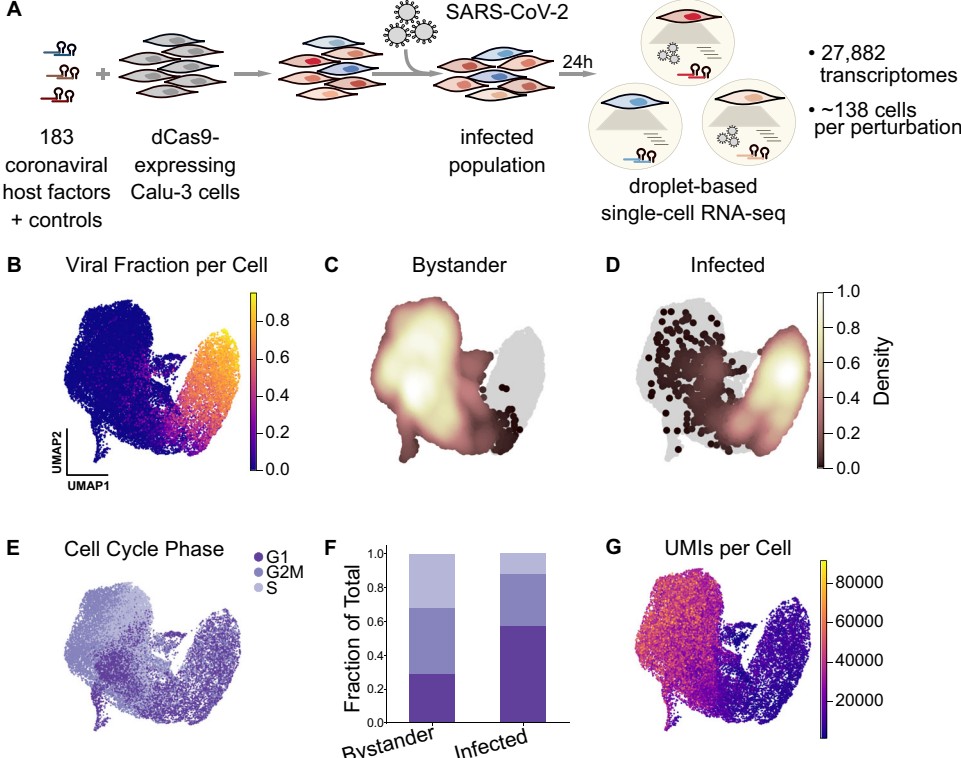

**Fig. 1 | Perturb-seq for single-cell transcriptional analysis and functional validation of SARS-CoV-2 host factors. A** Experimental design for the Perturb-seq experiment in Calu-3 cells engineered to express CRISPRi machinery. We perturbed 183 different host factors (individually or in combination) using a lentivirally-delivered library, infected the cells with SARS-CoV-2 for 24 hours, and performed droplet-based single-cell RNA sequencing, reading out host and viral transcripts as well as the sgRNA, indicating the perturbed host factor. **B** Single-cell transcriptomes were projected into UMAP space and colored by viral RNA fraction per cell. **C, D** Density of cells identified as either uninfected/bystander (**C**) or infected (**D**) by our classifier, overlaid onto all cells in gray. **E** Cells color-coded by their cell cycle phase. **F** Fraction of bystander and infected cells assigned to each cell cycle phase. **G** Cells color-coded by the number of detected UMIs per cell.

27,882 single cells with exactly one unambiguously assigned library element each.

## Transcriptional heterogeneity in SARS-CoV-2 infected cells

As a baseline for our subsequent Perturb-seq analysis, we first profiled the transcriptional response in the cell population upon infection, characterizing the spectrum of cellular states irrespective of the genetic perturbations present in the population. The heterogeneity of cellular states was primarily driven by the fraction of viral transcripts (Fig. 1B), which reached levels of up to 95% in some cells.

In order to compare infected and uninfected cells, we developed a classifier that determines the infection state of each cell based on the read counts of individual viral transcripts (Methods, Fig. 1C, D). Due to the presence of 'ambient' viral RNA, almost all cells have nonzero viral reads. To separate cells with true infection from those with spurious reads, the baseline of ambient viral RNA per cell was determined based on a spike-in of uninfected wild-type cells, which were identified by the absence of lentivirus-derived transcripts.

We sought to design an experimental strategy that captures single-cell transcriptomes of SARS-CoV-2 infected cells in a way that resolves both host and viral transcripts. Coronaviruses have a unique transcript architecture[39], consisting of the (+)strand viral genome, numerous subgenomic mRNAs (sgmRNAs), and matching (-) sense counterparts. Importantly, all (+)sense transcripts start with the same ~72nt leader sequence at the 5′ end, followed by a junction to the body of the sgmRNAs. All (+)sense transcripts also share the same 3′ end and are polyadenylated. We reasoned that 3′ sequencing would not be able to resolve individual viral transcripts and therefore used the 10x Genomics 5′ workflow with a modified sequencing strategy that extends read1 to sequence from the 5′ end into the transcript, spanning the leader-body junction (Supplementary Fig. 1A). A recent report found the same conceptual approach to maximize unambiguous detection of the different viral sgmRNAs[40].

Utilizing this 5′ sequencing strategy (Methods), we resolved individual viral sgmRNAs and observed distinct patterns of viral transcript abundances in infected cells (Supplementary Fig. 1B). The 3′-proximal Nucleocapsid (N) transcript was by far the most abundant viral RNA. Cell-by-cell correlation of the abundances of individual viral sgmRNAs was largely a function of genomic location: the abundances of the sgmRNAs proximal to N, encoding ORF3A, E, M, ORF6, ORF7ab, ORF8 showed the highest correlation with N. Conversely, the abundances of Spike and ORF1ab (i.e., whole genome) were much less correlated on a cell-by-cell basis. Additionally, we mapped the positions of leader-body junctions in sgmRNAs from our extended read1 data and found both the positions as well as their relative frequencies of individual junctions to be in agreement with measurements derived from bulk, whole-transcript sequencing data[39] (Supplementary Fig. 1C).

Next, using our infection state classification, we observed the cell cycle phase as a major contributor to the heterogeneity among uninfected cells, with subclusters often representing cells within one predominant phase. Conversely, infected cells showed a pronounced, general shift in their cell cycle phases: we observe far fewer infected cells in S phase and the proportion of G1 cells increased approximately two-fold (Fig. 1E, F), suggesting that cell cycle arrest occurs upon infection. Furthermore, infected and uninfected bystander cells differed dramatically in the total amount of detectable RNA per cell, quantified by the number of unique molecular identifiers (UMIs) (Fig. 1G, Supplementary Fig. 1D). We calculated Pearson's correlation coefficient between total UMIs per cell and viral fraction per cell, and observed a negative correlation ($r = -0.44$). These data indicate a pronounced shutoff of host gene expression in infected cells. This observation is consistent with a recent study showing that SARS-CoV-2 NSP1 specifically degrades transcripts lacking the viral 5′ leader sequence, enabling the virus to dominate the cellular mRNA pool[41].

To further characterize the heterogeneity within the infected and bystander populations, different cell states were delineated using Leiden clustering, defining 12 clusters of bystander cells (clusters A–L) and 7 clusters of infected cells (clusters M–S) (Fig. 2A).

To identify transcriptional patterns within these different clusters, we evaluated gene expression within each cluster. (Fig. 2B). Bystander cells (clusters A–L) varied in their expression of genes associated with antigen presentation, chemokines, and interferon-stimulated genes (ISGs). ISGs including *IFI6*, *IFI27*, and *ISG15* (Fig. 2C) were prominently more abundant in bystander cells compared to infected cells. This suggests active suppression of the interferon response in infected cells, a phenomenon that has been observed for many different viruses[31,33,41,42].

We identified a small but prominent subset of cells (bystander cluster L and infected cluster M) expressing interferon β (*IFNB1*) and λ (*IFNL1/2/3*) (Fig. 2D) and a number of chemokines (*CXCL1/2/3/10/11*, *CCL5*, *IL6*, *CXCL8/IL8*). This observation is consistent with prior single-cell work showing a subset of interferon-producing cells after SARS-CoV-2 infection[17], and studies that assessed interferon production in bulk[43]. Notably, all interferon-producing cells exhibited pronounced expression of both NF-κB pathway genes and ISGs. Additionally, this population expressed genes associated with antigen presentation and translation regulation/stress response (e.g., *PPP1R15A*). These features were reminiscent of subpopulations of abortively infected cells which have been characterized for the herpesviruses HSV-1 and HCMV[31,33]. However, only ~20% of the interferon-producing cells in our dataset were classified as infected based on the abundances of viral transcripts (cluster M).

Infected cells (clusters M–S) varied not only in their fractions of viral transcripts, but also showed a concomitant shift in cell cycle distribution (Fig. 2A), and subtle host transcriptional patterns (Fig. 2B). A number of host transcripts were generally upregulated in infected cells, including genes associated with NF-κB signaling such as *NFKBIA* (Fig. 2E), *NFKBIE/Z*, *EGR1*, *REL* and *RELB* (Fig. 2E). In addition, genes related to cell stress (*ATF3*, *FOS*, *JUN*) were upregulated in most infected clusters. It is conceivable that the apparent downregulation of some transcripts in infected cells (such as ISGs) is an artifact caused by the global host shutoff. Therefore, we repeated gene expression and cell cycle analyses on cells that were downsampled to the read depth of infected cells (bottom 2% of the UMI distribution). These data recapitulate our prior findings and suggest that despite host shutoff, we were able to detect transcriptional changes in infected and bystander cells (Supplementary Fig. 2A–F).

## Host perturbations alter infection dynamics

To determine how the activity of host factors affects the response of a cell population to SARS-CoV-2 infection, we next evaluated how each genetic perturbation in our CRISPRi library altered viral load and bystander activation. To ensure sufficient representation of our 239 library elements, we assessed the distribution of captured cells for these elements and determined the peak of that distribution (mode) to be at 138 cells (Supplementary Fig. 3A). 48 library elements had less than 55 cells each, forming a distinct lower mode in the distribution of cell numbers, suggesting that they target genes essential for the growth of Calu-3 cells. As these elements lacked appropriate coverage for proper evaluation of infection dynamics, they were removed, resulting in 25,835 remaining cells, on which we based all downstream analyses (Supplementary Data 2). Among well-represented targets, the median knockdown efficiency was 91%, and 80% of our library showed greater than 75% knockdown of their respective target transcripts in uninfected cells confirming the efficacy of CRISPRi targeting in Calu-3 cells (Supplementary Fig. 3B).

To test which host factors confer protection from infection upon perturbation, we compared the distributions of viral loads in cells with any given CRISPRi target against the population of cells with

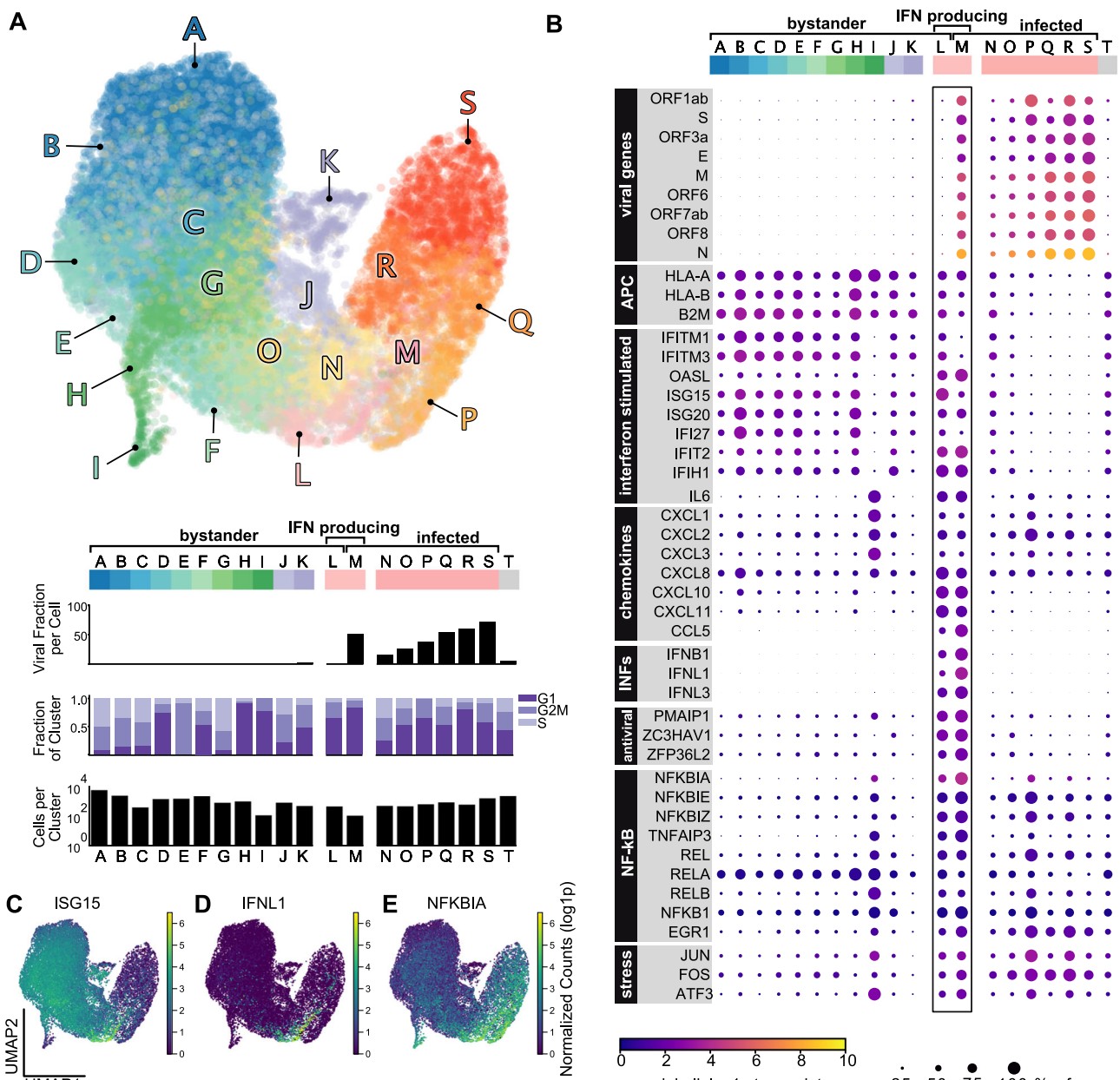

**Fig. 2 | Transcriptional heterogeneity in SARS-CoV-2 infection. A** Single-cell transcriptomes were projected in UMAP space and colored by Leiden cluster. Leiden clusters were subsequently characterized by the mean viral fraction, the number of cells, and the cell cycle composition per cluster. Cluster T are all cells that could not be assigned an unambiguous infection state. **B** Differential expression of Leiden clusters revealed transcriptionally distinct subclusters of bystander cells, infected cells, and a small subset of interferon-producing cells. The color of each dot is pseudobulk gene expression of each gene per cluster, and the size of each dot is the expression normalized to the cluster with maximum expression of that gene. **C–E** Host transcriptional analysis revealed heterogeneity in infected and bystander populations, including differential gene expression in UMAP space of: **C** ISG15; **D** IFNL1; and **E** NFKBIA.

non-targeting controls (Fig. 3A). Knockdown of only one factor, *SEC62*, resulted in increased viral loads. This was unexpected in light of genetic screens that identified *SEC62* knockout as protective against human coronavirus (HCoV) OC43 infection[9]. *SEC62* is involved in the post-translational targeting of proteins across the endoplasmic reticulum, acts as an autophagy receptor in the ER, and is a known interactor of *SEC61B*[44,45]. On the other hand, knockdown of the known entry factors *ACE2* and *TMPRSS2*, both alone and in combination, led to strongly reduced viral loads. Similarly, *TMPRSS2* in combination with either Furin, Cathepsin B, or L (but notably not Furin or either Cathepsin alone) resulted in substantially reduced fractions of viral RNA, suggesting partial redundancy of those entry factors. Knockdown of *BRD2* also reduced viral loads considerably, which is consistent with

the recent finding that *BRD2* is required for efficient transcription of *ACE2*[34].

Aside from those known factors involved in viral entry, we identified a number of additional, strongly protective factors such as the autophagy factor *ATG14*[46], as well as translation factors *EIF4E2* (*4EHP*) and *EIF4H*. Translation factors *EIF4E2* and *EIF4H* were previously found to interact with the viral proteins NSP2 and NSP9, respectively[1,2]. *EIF4E2* represses translation initiation by binding to the mRNA cap and can be ISGylated to enhance this cap-binding activity[47]. In the setting of SARS-CoV-2, *EIF4E2* surfaced as an unvalidated protective host factor in one genetic screen[5]. The second translation factor that conferred protection from infection upon knockdown, *EIF4H*, binds to and stimulates RNA helicase activity of *EIF4A*[48,49]. Additionally, *EIF4H* is reported to

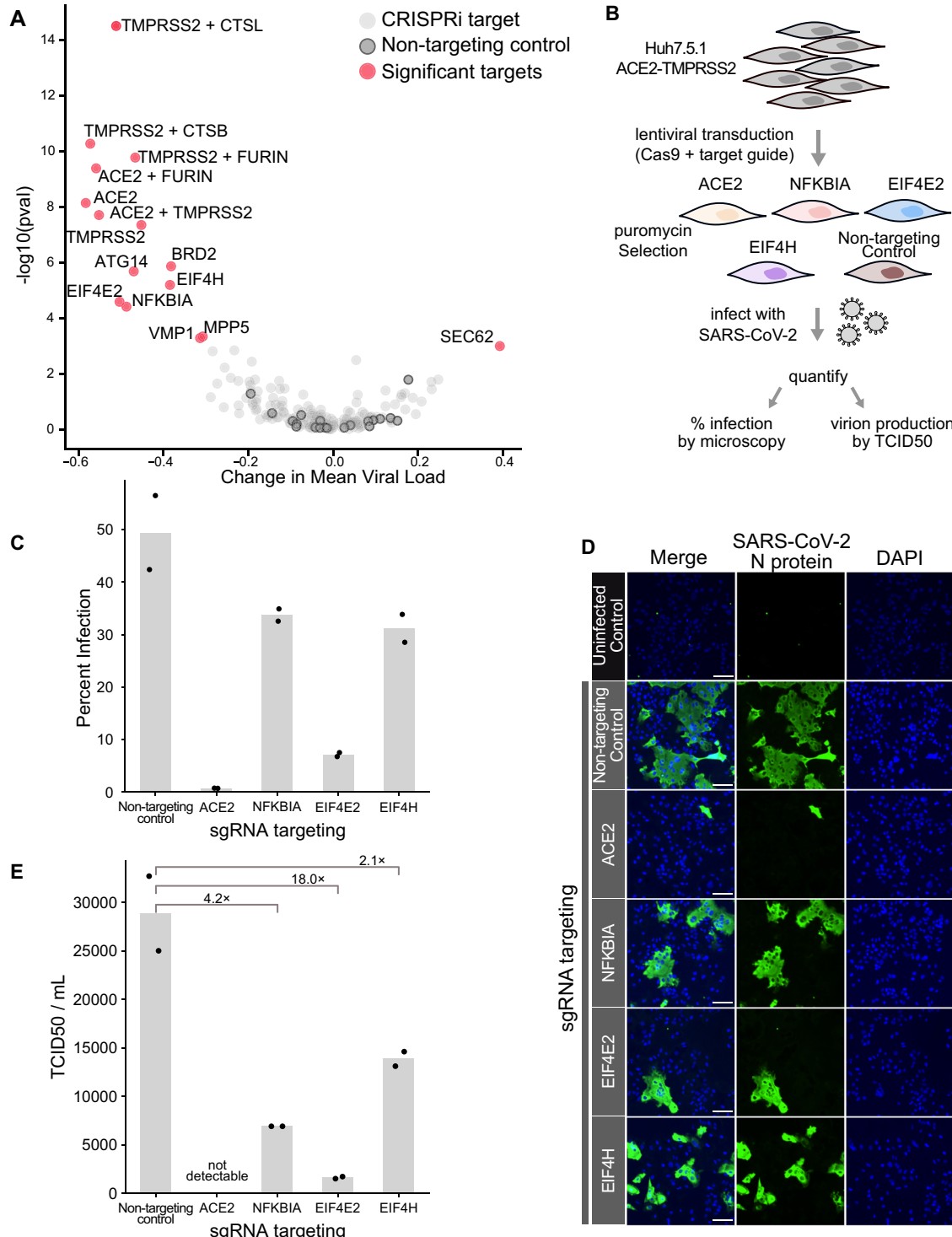

**Fig. 3 | Host perturbations alter SARS-CoV-2 infection dynamics. A** The effect of how each CRISPRi perturbation altered viral load was displayed as the change in mean viral load by KS p-value (Kolmogorov–Smirnov test, two-sided) of viral load distribution change. Color code indicates CRISPRi targets, non-targeting controls, and targets in which knockdown significantly altered viral loads. **B** To orthogonally validate CRISPRi targets, we transduced Huh7.5.1 cells overexpressing ACE2 and TMPRSS2 with lentivirus targeting control and test genes. Cells were subsequently infected with SARS-CoV-2, and percent infection was calculated (**C**) by immunofluorescence and microscopy (**D**), counting the fraction of cells positive for SARS-CoV-2 nucleoprotein. Original images were taken on an inverted fluorescence

microscope at 4× (>2000 cells). Representative, zoomed-in fields of view are shown and the scale bar represents 100 μm. In the bar plot, each bar represents the mean percent infection for a given cell line and points represent the individual data points. Additionally, we quantified infectious virion production using the TCID50 assay and calculated fold change for each cell line relative to the mean TCID50/mL of non-targeting control cells. **E** In the bar plot, each bar represents the mean TCID50/mL for a given cell line, and the points represent the individual data points. The ACE2 knockout positive control was measured once for the TCID50 assay; two biological replicates were performed for all other infection conditions (**C**–**E**). Source data for orthogonal validation are provided as a Source Data file.

interact with SARS-CoV-2 RNA[5]. Notably, the protective phenotypes when targeting *EIF4E2* and *EIF4H* do not appear to reflect a general effect of perturbing translation factors, as *EIF4B* did not significantly alter infection dynamics.

Additionally, knockdown of *VMP1* and *MPP5* conferred protection from infection. *VMP1* is involved in cytoplasmic vacuole formation and autophagosome assembly, when it interacts with *TMEM41B*[50], a known pan-coronavirus host dependency factor[5,9]. *MPP5* is involved in tight junction formation and was similarly identified as a protective hit in a genome-wide CRISPR survival screen[9], and as an interactor of the E protein of SARS-CoV-1[51]. Our data validate those proteins as protective host factors.

Lastly, we observed that knockdown of the NF-κB inhibitor IκBα (encoded by *NFKBIA*) significantly reduced viral loads. The NF-κB pathway is well-known to be activated in the setting of viral infections[52], and its activity was reported to be important for SARS-CoV-2 replication[53]. While *NFKBIA* is transcriptionally upregulated in SARS-CoV-2 infected cells as shown in our data (Fig. 2E) and by others[17,53,54], it has not appeared in any genetic screen to our knowledge as a protective factor.

At baseline, IκBα inhibits the NF-κB pathway by binding to and retaining p65/RELA-containing complexes in the cytosol[55]. Canonical pathway activation induces proteasomal degradation of IκBα/NFKBIA, leading to p65/RELA nuclear translocation and subsequent transcription of target genes (including NFKBIA, forming a negative feedback loop). Prior studies have shown that the papain-like proteases (PLPro) of both SARS-CoV-1 and SARS-CoV-2 can deubiquitylate and thereby stabilize IκBα, thus decreasing p65/RELA nuclear translocation and suppressing pathway activation[54,56]. However, our data show that knockdown of *NFKBIA* does not lead to transcriptional activation of the NF-κB pathway in bystander cells (Supplementary Fig. 4A), arguing against constitutive activation as a phenotypic outcome. Knocking down *RELA* or *RELB*, both individually or in combination, did not result in a protective phenotype. Our data suggest a dependency of SARS-CoV-2 on *NFKBIA*, which may be independent of its inhibitory role in the NF-κB pathway.

To further investigate the phenotypic response of *NFKBIA* perturbation, we utilized the OpenCell collection of HEK293T cell lines expressing split mNeonGreen (mNG)-tagged proteins from their endogenous loci[45]. First, we confirmed that the NF-κB pathway was functional in cells expressing mNG-tagged RELA. Using live-cell fluorescent microscopy, we observed the expected p65/RELA translocation to the nucleus after TNF-α stimulation (Supplementary Fig. 4B). We then generated polyclonal *NFKBIA* knockout lines in the background of the mNG-RELA line. Without stimulation, there was no constitutive p65/RELA translocation to the nucleus in *NFKBIA* KO cells. After treatment with TNF-α, we observed a blunted response with delayed and incomplete p65/RELA nuclear translocation in *NFKBIA* KO cells compared to control cells (Supplementary Fig. 4B). These data are in agreement with prior studies that show a delayed response to NF-κB pathway stimulation in the setting of an *NFKBIA* knockout[57], and suggest a compensatory mechanism that prevents both constitutive and acute pathway activation.

Next, we orthogonally validated the observed protective phenotypes of inactivated *NFKBIA*, *EIF4E2*, and *EIF4H* by generating knockout lines from Huh7.5.1 (hepatocellular carcinoma) cells ectopically expressing *ACE2* and *TMPRSS2* (Fig. 3B). This cell line is permissive for SARS-CoV-2 infection and has been used for pooled CRISPR screening[9]. First, we infected polyclonal pools of knockout cells with SARS-CoV-2 and quantified the fraction of infected cells by fluorescence microscopy, staining for the viral nucleocapsid protein. Compared to non-targeting controls, *ACE2*, *NFKBIA*, *EIF4E2*, and *EIF4H* knockout cell lines showed a substantial decrease in infection (Fig. 3C, D). *NFKBIA* knockout cells displayed a 31.8% decrease in infection, 85.5% in *EIF4E2* KO, and 33.2% in *EIF4H* KO cells compared to non-

targeting control cells. To determine if these perturbations altered SARS-CoV-2 production, we next quantified infectious virus production for each knockout line. At 24 h post infection, following similar trends to our viral intracellular staining, we saw a decrease in viral titer for these same knockout cells (Fig. 3E). Altogether, these data provide functional evidence that *NFKBIA*, *EIF4E2*, and *EIF4H* play a role in SARS-CoV-2 infection in multiple cell types, and suggest that this action is prior to viral transcription, viral translation, and egress.

## Systematic classification of host factor phenotypes

Changes in the viral load distribution are only one manifestation of the multitude of cellular phenotypes resulting from host factor perturbation. To achieve a systematic and unbiased characterization of host factor perturbation phenotypes, beyond viral protection/sensitization, we monitored how different perturbations shift the proportion of cells in distinct cellular states. Qualitatively, this can be assessed by looking at the distribution of cells with a given genetic perturbation on the UMAP projection. More quantitatively, one can count cells in the different Leiden clusters and determine how a given host factor perturbation changes the relative numbers of cells by cluster (see Fig. 2A). This approach not only identifies host factor perturbations that alter cellular states and sorts them by similarity, but also narrows down the underlying mechanism by directly pinpointing the cellular states that are affected by the perturbation[31].

Cells with non-targeting control sgRNAs were uniformly distributed across the UMAP representation of the cell population (Fig. 4A). In comparison, cells with certain genetic perturbations deviated from this pattern in specific ways. First, cells with sgRNAs targeting known entry factors were specifically excluded from all infected clusters in UMAP space (Fig. 4A, Supplementary Fig. 5A). The same was also true for cells with EIF4E2, EIF4H and NFKBIA-targeting sgRNAs (Supplementary Fig. 3C, Fig. 4A). Moreover, cells with sgRNAs targeting proviral factors were similarly depleted from two clusters (bystander clusters F and L, the latter being one of the interferon-producing clusters). These clusters border infected clusters and were classified as uninfected based on viral transcripts in quantities below noise level. Based on the observation that entry factor inactivation excludes cells from these clusters, we speculate that these two clusters represent cells that are in the earliest stage of infection, have been infected with a defective viral particle, or are in a state where transcription of viral genes is effectively suppressed by an antiviral host response.

We systematically quantified the under/overrepresentation of cells with a given host factor perturbation in individual clusters (Fig. 4B), compared to cells with non-targeting control sgRNAs. The results can be visualized as a heatmap of the odds-ratio of how targeting a certain host factor changes the occupancy of each cluster (Fig. 4C, Supplementary Fig. 5A), which can further be projected onto a UMAP of host factor phenotypes (Fig. 4D, Supplementary Fig. 5B).

This analysis re-confirmed the group of proviral factors, which are strongly protective when inactivated (Fig. 4D, Supplementary Fig. 5B, blue highlight). A second group of perturbations that caused a distinct re-distribution of cells across the individual clusters were cells with inactivated members of the interferon pathway (Fig. 4A, D, Supplementary Figs. 3C, 5B, orange highlight). Those cells were shifted from bystander clusters representing cells with high expression of ISGs (clusters B, D, E, F) to the cluster with a low degree of interferon response (cluster A) (see Supplementary Fig. 5A, Fig. 2B).

The group of interferon signaling factors contained not only expected genes (*IFNAR2*, *STAT2*, *IRF3*, *IRF9*), but also genes not routinely implicated in the interferon response such as *SPNS1*, *KEAP1* and *GPR89A/B*. To evaluate these in more detail, we scored the extent of interferon response in single cells based on a previously established list of ISGs that are readily detected by single-cell RNA sequencing[31]. We subsequently tested for statistically significant shifts in this

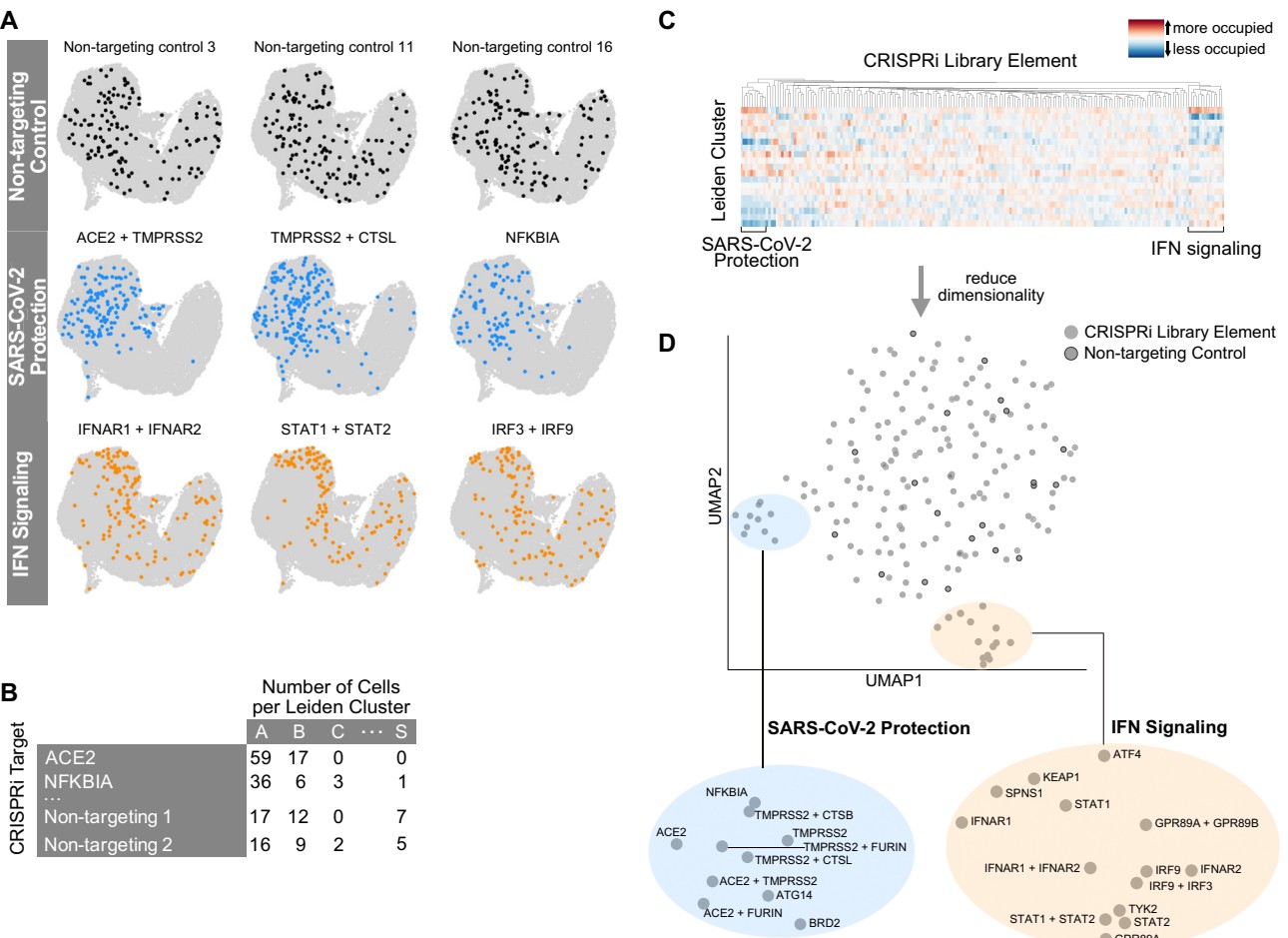

**Fig. 4 | Host perturbations alter localization of cells in UMAP space and Leiden cluster membership. A** Library elements for non-targeting controls, factors that alter SARS-CoV-2 infection, and interferon signaling are highlighted in UMAP space. **B** Library element representation by cluster was calculated, normalized, and visualized on a heatmap. **C, D** Subsequent dimensionality reduction of this odds-ratio was projected into UMAP space and revealed subclusters by biological function.

interferon score for each perturbation compared to non-targeting controls (Fig. 5A, B). To rule out the effect of viral antagonism of this pathway, we limited this analysis to bystander cells.

Knockdown of *GPR89A/B*, *KEAP1*, *SPNS1,* and *BRD2* significantly decreased bystander activation as measured by our ISG scores, confirming these proteins as regulators of the interferon signaling pathway. *GPR89A* and *GPR89B* are sequence-identical paralogs, encoding a G protein-coupled receptor, and proteomic studies report interactions of this protein with multiple SARS-CoV-2 proteins (M, NSP6, and ORF7B)[4]. Notably, *GPR89A/B* overexpression is reported to activate the NF-κB signaling pathway[58], and this protein is thought to be important for Golgi acidification and glycosylation[59]. *KEAP1* is a repressor of *NRF2*, which acts as a regulator of the inflammatory response[60]. Our findings for *KEAP1* are consistent with prior work that showed repression of inflammatory genes in *Keap1* deficient murine cells[61]. *SPNS1* is involved in lipid and transmembrane transport, and Wang et al. reported that genetic knockout of this gene protects from hCoV-229E and hCoVOC43 infections in vitro[9]. While both *KEAP1* and *SPNS1* were shown to interact with SARS-CoV-2 ORF3 and ORF7b, respectively[1,4], we only saw an effect on bystander activation in these experiments. Furthermore, CRISPRi knockdown of *BRD2* decreased the overall sensitivity of bystander cells in our study, which aligns with prior reports that perturbation of *BRD2* reduces interferon signaling[34]. Taken together, our analytical framework identified genes not routinely implicated in bystander activation, and proved to be very sensitive to identify factors with subtle phenotypes beyond strong protection from infection.

## Discussion

In this study, we measured the dynamics of SARS-CoV-2 infection in tissue culture, and simultaneously validated and functionally characterized host factors of infection. Perturb-seq delivers a high-dimensional, phenotypic single-cell readout, characterizing both the intrinsic heterogeneity of a SARS-CoV-2 infected population, and the response to many host factor perturbations. We captured different functional outcomes and simultaneously classified host factors by the similarities of their roles during infection and bystander activation. Our study thereby complements and greatly expands upon the genomic and proteomic screens which initially informed our selection of host factors included in our Perturb-seq library[1,2,4–6].

Our transcriptional analysis revealed upregulation of key NF-κB pathway members, including *NFKBIA*, in SARS-CoV-2 infected cells. This is consistent with findings of earlier studies[17,18,53]. Considering the prominent transcriptional host shutoff, we speculate that viral factors trigger the upregulation of *NFKBIA* and/or protect the transcript from degradation. Moreover, our study demonstrates that IκBα/*NFKBIA* can be targeted genetically to confer strong protection from SARS-CoV2 infection. Collectively, we show that perturbation of *NFKBIA* decreased viral RNA production, viral protein production, and the production of infectious progeny, suggesting its necessity for completion of the viral life cycle.

Furthermore, our data suggest blunting of NF-κB pathway activation as one underlying mechanism to explain this phenotype. While somewhat counterintuitive in light of IκBα/*NFKBIA*'s role as an NF-κB

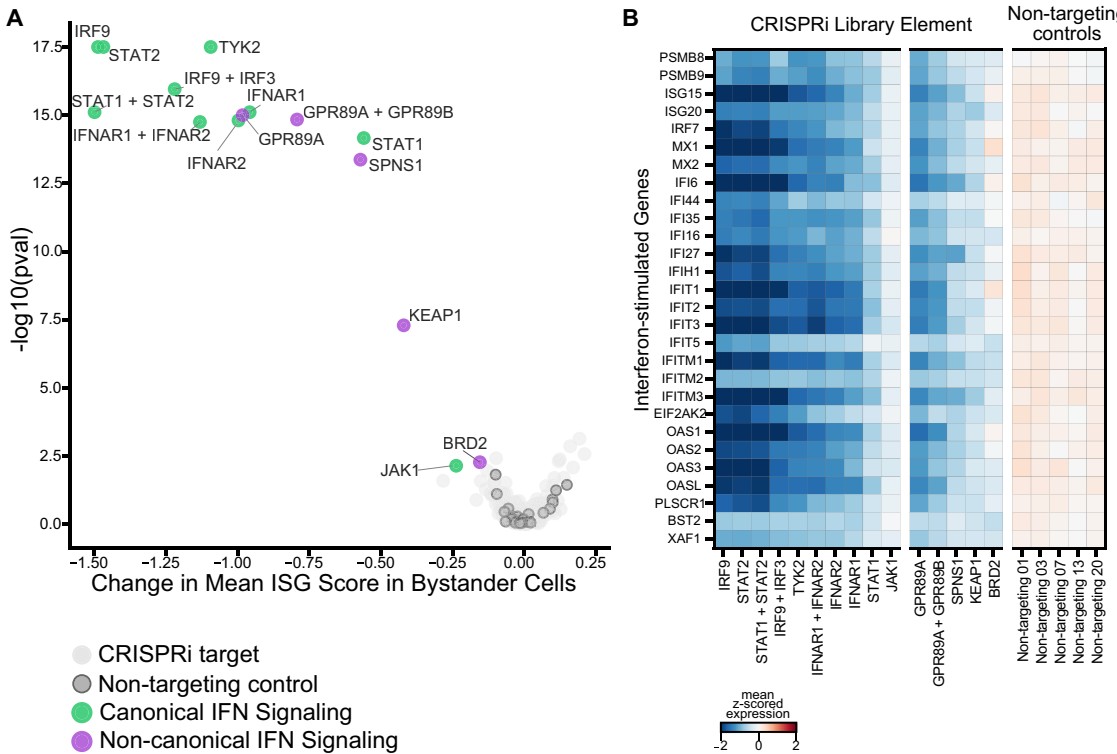

**Fig. 5 | Host perturbations alter interferon signaling in bystander-activated cells. A** We scored bystander cells based on their ability to respond to interferon (ISG score) and tested which perturbations significantly altered the ISG score distribution by perturbation. This is represented by the mean change in ISG score when compared to non-targeting controls by the KS *p* value (Kolmogorov–Smirnov test, two-sided) per perturbation. **B** Expression heatmap of select targeting and non-targeting library elements showing the mean z-score for a subset of interferon-stimulated genes.

inhibitor, this result is in line with data from a previous optical imaging screen that shows a p65 translocation defect upon *NFKBIA* perturbation[57]. Furthermore, we suspect that independent from IκBα's canonical inhibitory role in NF-κB signaling, IκBα may additionally be co-opted in another way that benefits viral proliferation. This is corroborated by a recent report that overexpression of a dominant-negative IκBα mutant enhances SARS-CoV-2 infection in A549 cells, while simultaneously reducing p65/RELA nuclear translocation[62]. These results, together with our transcriptional and translocation data, underscore the importance of IκBα during the SARS-CoV-2 life cycle and suggest its role may be independent of activation of the NF-κB pathway.

Our data further establishes that two translation factors, *EIF4E2* and *EIF4H*, are required for SARS-CoV-2 infection. While prior studies report that both factors interact with viral proteins[1], here, we show that knockdown and knockout of these factors decrease infection. The 4EHP(EIF4E2)-GIGYF2 complex is involved in ribosome-associated quality control by preventing translation initiation of faulty mRNA[63–65], and its interaction with NSP2 is conserved across SARS-CoV-1, SARS-CoV-2 and MERS-CoV[2]. Others have proposed that viral NSP2 interacts with the 4EHP(EIF4E2)-GIGYF2 complex to inhibit host translation initiation[66]. However, the strongly protective knockdown phenotype of *EIF4E2* observed in our data leads us to instead hypothesize that binding of viral NSP2 to EIF4E2 drives preferential translation of viral RNA. In this manner, the virus may subvert what is normally a defense mechanism for its exclusive use within the cell. Further investigation to determine which transcripts EIF4E2 binds to in the setting of infection with ribosome profiling will aid our understanding of the underlying mechanism of *EIF4E2* utilization by coronaviruses.

EIF4H directly binds to and stimulates the DEAD box RNA helicase EIF4A[49]. A pharmacological inhibitor of EIF4A, Zotatafin, decreases SARS-CoV-2 infection in vitro, and clinical trials (NCT04632381) are

underway to evaluate its safety and efficacy in humans[1,67]. Our experiments reveal a viral dependency on the EIF4A binding partner EIF4H, suggesting a complementary, and possibly synergistic point for additional therapeutic intervention.

In addition to characterizing the consequences of inactivation of proviral factors during SARS-CoV-2 infection, Perturb-seq enabled us to identify *SEC62* as an antiviral factor. Contrary to our initial hypothesis that similar to OC43, *SEC62* knockdown would provide protection from SARS-CoV-2 infection, we instead observed sensitization. While this diverges from OC43, a similar infection enhancement has been observed with *SEC62* knockdown and Foot-and-mouth disease virus[68]. Moreover, *SEC62* is a dependency factor for HIV replication, and notably, knockdown alters cell-surface expression of specific trans-membrane proteins necessary for HIV infection[69]. Our study adds to mounting evidence that *SEC62* is important for viral infections, but further investigation is warranted to interrogate if the mechanism behind our SARS-CoV-2 finding is due to modulation of autophagy, ER stress, and/or transmembrane protein translocation for cell-surface expression. Finally, our systematic characterization of each genetic perturbation revealed regulators of bystander activation. *KEAP1*, *GPR89A/B*, and *SPNS1*, which were previously found to be protective when knocked out[5,7,9], did not alter infection dynamics within our study. We speculate that knockout of these genes was identified as protective in survival screens due to their lack of interferon sensitivity, leading to protection from interferon-induced death[5,9]. Conversely, it is possible that these contrary phenotypes are representative of the different timeframes of our Perturb-seq experiments (24 hours) compared to genetic survival screens (7+ days). While knockdown of *KEAP1*, *GPR89A/B*, and *SPNS1* initially decreased interferon stimulation in our experiments, it is conceivable that these factors have a secondary role in protecting the population from infection in long-term cultures. To further investigate how these factors alter infection

dynamics and the innate immune response over the course of infection, experimentation in different models of the respiratory epithelium is warranted.

While our Perturb-seq library was designed to include genes with experimental evidence of roles in coronavirus biology, only ~13% of these factors ultimately showed significant phenotypes during the first 24 hours of infection in our cell culture model. This underscores the necessity for high-throughput orthogonal validation and characterization of host factors in different cell types. We do expect that specific host factor perturbation phenotypes, in particular of factors acting at the later stages of the viral life cycle such as virion assembly and egress, cannot be resolved by Perturb-seq. Similarly, host factors that are active only in rare subsets of cells, such as the interferon-producing subpopulation, may be difficult for Perturb-seq to dissect without increasing the scale of these experiments.

In summary, our study presents comprehensive transcriptional profiling of SARS-CoV-2 infection dynamics, tests the effect of 183 host factor perturbations on infection, and characterizes the host response of each perturbation. Key advances of this work include the identification of genes involved in bystander activation and functional validation of host dependencies factors of SARS-CoV-2. Our study highlights the utility of Perturb-seq for large-scale systematic characterization of host factors essential for pathogen infections and establishes the groundwork for future mechanistic studies to investigate how SARS-CoV-2 modulates both the NF-κB pathway and translation.

## Methods

### Establishment and propagation of SARS-CoV-2 clinical isolate

SARS-CoV-2 (SARS-CoV-2/human/USA/CA-UCSF-0001H/2020) was isolated, propagated, and plaqued on Huh7.5.1 cells overexpressing ACE2 and TMPRSS2[9]. Viral titer was determined using standard plaque assay with Avicel[70] on Huh7.5.1-ACE2-TMPRSS2 cells. Isolated virus was sequence-verified, lineage identified using PANGO[71], and deposited onto GISAID (accession ID: EPI_ISL_13689582). Additionally, SARS-CoV-2 was mycoplasma negative (Lonza MycoAlert Mycoplasma Detection Kit). All experiments in this study that utilized cultured SARS-CoV-2 were conducted in a biosafety-level 3 laboratory.

### Cell culture

The CRISPRi Calu-3 cell line was generated by lentiviral delivery of pMH0001 (UCOE-SFFV-dCas9-BFP-KRAB)[19] (Addgene #85969) into Calu-3 cells, followed by FACS sorting of BFP positive cells[19,34]. These cells were grown in DMEM/F12 supplemented with 10% FCS, penicillin, streptomycin, glutamine, and non-essential amino acids. Huh7.5.1 cells overexpressing ACE2-TMPRSS2 and HEK293T cells were grown in DMEM supplemented with 10% FCS, penicillin, streptomycin, and glutamine. All cell types were maintained at 37 °C and 5% CO2. Calu-3 and HEK293T cells were obtained from the UCSF Cell and Genome core. Huh7.5.1 cells overexpressing ACE2-TMPRSS2 were obtained from Andreas Puschnik.

### Library design and lentivirus generation

Our Perturb-seq library was designed to target coronavirus host factors which were compiled from the literature, primarily from proteins physically interacting with coronavirus proteins, and from genes that came up as hits in CRISPR screens for host factors. All targets, sgRNA sequences, and host factor annotations are listed in Supplementary Data 1. Guide selection and library cloning followed the design introduced by Replogle et al.[36,37]. We used a lentiviral backbone (pJR101, a variant of pJR85, Addgene #140095, with a GFP instead of BFP marker) which carries an additional Puromycin marker and allows the expression of two separate sgRNAs from different U6 promoters (human and mouse, respectively) with two distinct sgRNA constant regions (CR1 and CR3, respectively) to remove homologous regions in order to

minimize recombination during lentiviral packaging. CR1/3 are further engineered with 'capture sequence 1' to be compatible with 10x's direct guide capture technology of the non-polyadenylated sgRNAs[36]. Guide oligos containing sets of two sgRNA sequences, separated by a spacer region, were ordered from Twist Bioscience, PCR-amplified, and cloned into pJR101 by ligation into the BstXI/BlpI restriction sites. The BsmBI-flanked spacer was then replaced by a fragment amplified from pJR98 (Addgene #140096), carrying the constant region of the first sgRNA and the promoter for the second one. The resulting library was sequenced to confirm proper guide sequences and abundance distribution.

After initial library cloning was completed, we obtained new screening data and designed an additional 24 sgRNAs, targeting 12 factors with 2 sgRNAs each. Those were cloned in an array into the same pJR101 background as one-guide vectors (without the pJR89 drop-in). We then pooled the individually cloned sgRNA vectors with the initial library at equimolar amounts of all library elements at the DNA level. We used this combined library for lentiviral production as described[36]. While analyzing our single-cell datasets, we observed that the individually cloned library elements were overrepresented roughly 3-fold, which we attribute to higher lentiviral packaging efficiency due to their slightly smaller size.

### Perturb-seq

Calu-3 CRISPRi cells were transduced with our Perturb-seq library at an MOI of ~0.1. Cells were puromycin-selected for 7 days, after which they had plateaued at ~93% GFP+ cells, followed by two more days of culture without selection markers. Cells were seeded into a 12-well plate at 400,000 cells/well and on the following day infected with SARS-CoV-2 at an MOI of 4. Infection was performed either for 1 h, followed by a media change ('pulsed infection') or without removal of the inoculum ('non-pulse'). After 24 h, cells were washed with PBS, dissociated with TrypLE Select Enzyme (10x, Thermofisher Scientific), washed, and resuspended in 1× PBS with 0.04% BSA. Wild-type, uninfected Calu-3 cells were spiked at ~1% into the dissociated Calu-3 CRISPRi cells to allow for analysis of ambient viral RNA. Manufacturer's instructions for the Chromium NextGEM Single-Cell V(D)J Reagents Kit v1.1 (10x Genomics) were followed for preparation of gene expression libraries. Modifications to the 10x single-cell sequencing protocol were made for direct guide capturing and library preparation as previously described[36]. Gene expression and guide libraries were subsequently quantified on the Bioanalyer (Agilent) using the High Sensitivity DNA kit, pooled, and sequenced on the Illumina NovaSeq 6000 (read1: 150 bp, read2: 150 bp, index length: 8 bp).

### Data analysis

Gene expression libraries were aligned using the 10x Genomics CellRanger v3.1.0 with default settings and aligned the hg38 reference genome concatenated with the SARS-CoV-2 genome. For viral alignments, STARsolo (version 2.7.8a) was used to capture viral junction sites. Cell barcode and UMI were identified for guide libraries using CellRanger. Guides identity was assigned to single cells following the Replogle et al. mixed model approach. Infection conditions were combined for downstream analyses since there was not a statistically significant distribution in guides between the conditions.

Scanpy was used for downstream cell filtering and analyses[72]. Cell filtering was done to include only cells that have one-guide set per cell and at least 55 cells per guide. Additionally, low-quality cells characterized as the bottom 2% of cells in total counts and cells with greater than 20% mitochondrial RNA were excluded.

We found the two populations with pulsed vs non-pulsed infection to exhibit very similar characteristics and combined them for all downstream bioinformatic analyses (Supplementary Fig. 2G). For assessing the effect of the host transcriptome in the setting of vastly different library distributions, we performed experiments with and

without viral sequences, and subsequently downsampled to the level of infected cells and re-analyzed the data.

For clear identification of infected cells, ambient viral RNA was evaluated in wild-type Calu-3 cells packaged into droplets. These WT control cells were identified by selecting cells that lack Cas9, lentiviral, and guide transcripts. We additionally selected cells that have at least 10,000 UMIs, which yielded 1082 cells for this analysis. In those cells, we determined the mean and standard deviation of the read counts of all individual viral genes. Other cells were considered infected if they had at least 6 viral transcripts at 2 standard deviations above the mean of WT cells, as well as more than twice the total viral transcript reads per cell. Conversely, cells were considered uninfected if no viral gene exceeded the 2 standard deviation threshold. A small proportion of cells could not be clearly determined as infected or uninfected, therefore we classified these cells as the borderline population (designated cluster T in UMAP space).

Guide knockdown percentages were determined by calculating the normalized count of target gene/non-targeting control. This analysis was limited to bystander cells to remove the effects of viral antagonism, and was subset to genes with at least 0.5 UMI per cell after normalization to remove low abundance or undetectable genes.

Cell cycle phases were determined following scanpy's tutorial. Similarly, single cells were also scored for interferon stimulation (ISG score) using scanpy's sc.tl.score_gene function. Differential expression was performed by exporting scanpy's count matrix to R, and subsequently performing MAST following Seurat's tutorials[73].

## Orthogonal validation

For targeted follow-up, published protocols for guide design and cloning into the lentiCRISPR v2 plasmid were followed[74]. The following sgRNA sequences were used:

ACE2: CAGGATCCTTATGTGCACAA;

NFKBIA: AGGCTAAGTGTAGACACGTG (Huh7.5.1), CTGGACGACCGCCACGACAG (HEK293T);

EIF4H: CCCCCCTACACAGCATACGT; EIF4E2: TCATAGCTCTGTGAGCTCGT.

Lentivirus was produced in HEK293Ts by co-transfecting pMD2.G, DR8.91, and the lentiCRISPR v2 plasmid with the guide of interest using TransIT-Lenti (Mirus Bio). Lentivirus-containing supernatant was collected 48 hours after transfection, filtered, and frozen.

For orthogonal validation of host factors that alter viral infection, Huh7.5.1-ACE2-TMPRSS2 cells were transduced with lentivirus in the presence of polybrene. We next selected transduced cells for 72 h with puromycin. After 1 week, knockout cell lines were infected with SARS-CoV-2 at an MOI of 3 for 20 hours in biological duplicates. Cells were subsequently fixed in 4% paraformaldehyde for 30 minutes, permeabilized with 0.2% Triton X, blocked with 5% BSA stained with primary anti-NP (Sino Biological 40143-R001), and secondary goat anti-rabbit IgG conjugated to Alexa Fluor 488 (Thermo Fisher Scientific A-11034). Slides were mounted with DAPI Fluoromount-G (SouthernBiotech 0100-20) and imaged on a Nikon Ti inverted fluorescence microscope (×4). Quantification of images was performed using CellProfiler 4[75].

To quantify infectious virus produced from our Huh7.5.1-ACE2-TMPRSS2 CRISPR KO lines, we seeded cell lines in duplicate and infected with SARS-CoV-2 at an MOI 0.05 at 37 °C. Viral inoculum was removed after 1 hour. At 24 h post infection, supernatant was collected, spun to remove cellular debris, and frozen at −80 °C. The infectious titer produced by each cell line was determined using the tissue culture infectious dose (TCID50) assay. Briefly, supernatant for each cell line was serially diluted and each dilution added to 10 wells of Huh7.5.1-ACE2-TMPRSS2 cells. After 6 days, CPE was determined by microscopy, and the TCID50/mL calculated using the Reed-Muench formula.

To investigate the effect of NFKBIA knockout on NF-κB induction, we generated NFKBIA knockout lines and controls in a background of HEK293T cells expressing N-terminally mNG11-tagged RELA[45]. RELA-tagged cell lines were transduced with lentivirus carrying Cas9 and NFKBIA-targeting sgRNA and puromycin-selected for 1 week. Cells were stimulated with recombinant TNF-α (50 ng/ml; Abcam ab9642) and imaged using confocal microscopy 25 and 45 minutes after stimulation. The imaging volume per field of view was 21 μm depth with 0.25 μm z-sectioning. During imaging, cells were maintained in a stage-top incubator (Okolab, H201-K-Frame) at 37 °C and 5% CO2. The imaging was performed using a DMI-8 inverted microscope (Leica) with a Dragonfly spinning-disk confocal (Andor) with a ×63 1.47 NA oil objective (Leica). Images were acquired using a Prime BSI sCMOS camera (Photometrics, pixel size = 6.5 μm × μm). Microscope control was achieved with Micromanager version 2.0.0[76]. Image visualization was via napari v0.4.16[77].

## Reporting summary

Further information on research design is available in the Nature Portfolio Reporting Summary linked to this article.

## Data availability

Raw and preprocessed data are available on GEO (GSE208240). Source data are provided with this paper.

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

## Acknowledgements

We would like to thank Melanie Ott, Camille Simoneau, Madhura Raghavan, Matthew Zinter, Cristina Tato, Charles Langelier, Ashley Byrne, Sandra Schmid, Amy Kistler and Carolina Arias for helpful discussions; Raul Andino for BSL-3 laboratory support; Kari Herrington and the Nikon Imaging Center at UCSF for microscopy support; Angela Detweiler and Norma Neff for sequencing support, and Verina Todorova for assistance. We additionally express our gratitude to the DeRisi and Weissman labs for helpful discussions and feedback. This work was supported by the Chan Zuckerberg Biohub (J.L.D., A.S.P., M.Y.H.), Howard Hughes Medical Institute (J.S.W., M.Y.H.), the National Institute of Health (F31AI150007 to S.S.) and Defense Advanced Research Projects Agency (DARPA PREPARE, HR0011-19-2-2007 to J.S.W.). This work does not necessarily represent the official views of the NIH.

## Author contributions

S.S., J.L.D. J.S.W. and M.Y.H. conceptualized the study. M.Y.H., S.S., A.S.P. and J.M.R. designed the Perturb-seq library with input from M.B.F. A.S.P. and M.Y.H. cloned the library. J.K.N. engineered Calu-3 CRISPRi cells. S.S. performed live-virus, single-cell, and follow-up experiments with help from M.Y.H., B.S.Z., J.L. and M.T.L. O.S.R. oversaw BSL-3 work. M.D.L. provided the mNG-RELA cell line and microscopy infrastructure. S.S., M.Y.H., J.R.B. and A.H.M. performed microscopy follow-up work on RELA translocation. S.S. and M.Y.H. analyzed the data with input from X.Q., C.J.Y. and J.W. S.S. and M.Y.H. wrote the manuscript, and J.L.D., J.S.W. and J.W. edited the manuscript. All authors commented on the manuscript.

## Competing interests

J.L.D. is a paid scientific advisor for Allen & Co. J.L.D. is a paid scientific advisor for the Public Health Company, Inc. and holds stock options. J.L.D. is a founder and holds stock options for VeriPhi Health, Inc. J.S.W. declares outside interest in 5 AM Ventures, Amgen, Chroma Medicine, KSQ Therapeutics, Maze Therapeutics, Tenaya Therapeutics, Tessera Therapeutics, and Third Rock Ventures. J.S.W. is an inventor on US Patent 11,254,933, related to CRISPRi screening, and has filed patents related to Perturb-seq that do not restrict academic use. M.Y.H. is a consultant for Illumina, Inc. J.M.R. is a consultant for Maze Therapeutics and Waypoint Bio. The remaining authors declare no competing interests.

## Additional information

[1]Department of Biochemistry and Biophysics, University of California San Francisco, San Francisco, CA, USA. [2]Chan Zuckerberg Biohub, San Francisco, San Francisco, CA, USA. [3]Department of Cellular and Molecular Pharmacology, University of California, San Francisco, San Francisco, CA, USA. [4]Whitehead Institute for Biomedical Research, Cambridge, MA, USA. [5]University of California, Berkeley-UCSF Joint Graduate Program in Bioengineering, San Francisco,

CA, USA. [6]Department of Medicine, Division of Pulmonary, Critical Care, Allergy and Sleep Medicine, University of California, San Francisco, San Francisco, CA, USA. [7]Department of Molecular & Cell Biology, University of California, Berkeley, Berkeley, CA, USA. [8]Department of Microbiology and Immunology, Center for Pathogen Research, University of Maryland School of Medicine, Baltimore, MD, USA. [9]Department of Cell and Tissue Biology, University of California, San Francisco, San Francisco, CA, USA. [10]Center for Cancer Research, Medical University of Vienna, Vienna, Austria. [11]Howard Hughes Medical Institute, Massachusetts Institute of Technology, Cambridge, MA, USA. [12]Department of Microbiology and Immunology, University of California, San Francisco, San Francisco, CA, USA. [13]Division of Rheumatology, Department of Medicine, University of California, San Francisco, San Francisco, CA, USA. [14]Institute of Human Genetics, University of California San Francisco, San Francisco, CA, USA. [15]Department of Epidemiology and Biostatistics, University of California, San Francisco, San Francisco, CA, USA. [16]Bakar Computational Health Sciences Institute, University of California, San Francisco, San Francisco, CA, USA. [17]Parker Institute for Cancer Immunotherapy, San Francisco, CA, USA. [18]Department of Biology, Massachusetts Institute of Technology, Cambridge, MA, USA. [19]David H. Koch Institute for Integrative Cancer Research, Massachusetts Institute of Technology, Cambridge, MA, USA. [20]Howard Hughes Medical Institute, University of California, San Francisco, San Francisco, CA, USA. [21]Max Perutz Labs, Vienna Biocenter Campus (VBC), Vienna, Austria. [22]Medical University of Vienna, Center for Medical Biochemistry, Vienna, Austria. ✉e-mail: weissman@wi.mit.edu; joe@derisilab.ucsf.edu; marco.hein@maxperutzlabs.ac.at

