## [Peer review file · Nature Communications]

REVIEWERS' COMMENTS

Reviewer #2 (Remarks to the Author):

Sunshine and collaborators perform an in-depth characterization of the host factors involved in SARS-CoV-2 infection using Perturb-seq, providing a valuable and detailed dataset to understand SARS-CoV-2 infection. The additional functional validation experiments that the authors performed greatly improved the manuscript and strengthened the Perturb-seq findings.

Minor comments:

- Figure 1 C&D – Please label the figure legend (i.e. “Density”).
- Figure S2 A-C – They are relevant for interpreting the layout, maybe consider moving to the main figure.
- Figure S4B is relevant for the understanding of the NFKBIA function, consider including it in the main figure.
- Figure 4A – Please include the localizations of several other key library elements (EIF4E2, EIF4H for “SARS-CoV-2 protection”, and KEAP1/SPNS1/GPR89 for “IFN signaling”).
- Figure 4D – Although with greater significances than NFKBIA in Fig 3A, neither EIF4E2 nor EIF4H was listed in Figure 4D. Were they not near the “SARS-CoV-2 protection” cluster? If so, please discuss the biological relevance.
- Figure 4B is a demonstration of how the authors generated Figure 4C, seems redundant, consider moving it as a supplemental table.
- Figure 5B, the legend indicates the z-score from dark blue to dark red, but in the figure only light red are shown. I understand the authors showed selected library elements, but are there any knockdowns that consistently increased overall ISG expression?
- One of the main findings in this paper is the dependency on NFKBIA by SARS-CoV-2. The authors and Simoneau et al. demonstrated such dependency is likely the result of disruption on the NFkB nuclear translocation, highlighting a nonlinear feedback loop by NFKBIA. Besides NFKBIA, TNFAIP3 (A20) is also well-known inhibitor of NFkB activation, was it one of the Perturb-seq library targets? If so, does TNFAIP3 knockdown also results in similar phenotype?

Reviewer #3 (Remarks to the Author):

The authors have sufficiently addressed all concerns.

Reviewer #2

Sunshine and collaborators perform an in-depth characterization of the host factors involved in SARS-CoV-2 infection using Perturb-seq, providing a valuable and detailed dataset to understand SARS-CoV-2 infection. The additional functional validation experiments that the authors performed greatly improved the manuscript and strengthened the Perturb-seq findings.

Response:

We thank the reviewer for all their constructive input and their positive assessment of our revised manuscript!

Minor comments:

-Figure 1 C&D – Please label the figure legend (i.e. “Density”).

Response: We have added “density” to the color scale.

-Figure S2 A-C – They are relevant for interpreting the layout, maybe consider moving to the main figure.

Response: We appreciate this suggestion and have added these 3 UMAPs to main Figure 2 . The manuscript text, figures and figure legends have been updated accordingly.

-Figure S4B is relevant for the understanding of the NFKBIA function, consider including it in the main figure.

Response: We agree that this is relevant for understanding the function of NFKBIA, but we feel this is better suited for a supplemental figure to avoid confusion about methods and cell-types used for these specific experiments.

-Figure 4A – Please include the localizations of several other key library elements (EIF4E2, EIF4H for “SARS-CoV-2 protection”, and KEAP1/SPNS1/GPR89 for “IFN signaling”).

Response: We thank the reviewer for this suggestion and have added the localization of these key library elements to Figure S3. The manuscript text, figure and figure legend has been updated accordingly.

-Figure 4D – Although with greater significances than NFKBIA in Fig 3A, neither EIF4E2 nor EIF4H was listed in Figure 4D. Were they not near the “SARS-CoV-2 protection” cluster? If so, please discuss the biological relevance.

Response: These factors were located at the very edge of the “SARS-CoV-2 protection” cluster and we did not label them in the main figure to avoid clutter. Their positions proximal to the cluster are apparent in Supplementary Fig. 5B, which is the high-resolution version of Fig. 4D with a complete set of labels.

-Figure 4B is a demonstration of how the authors generated Figure 4C, seems redundant, consider moving it as a supplemental table.

Response: We agree that the purpose of Fig. 4B is mainly to illustrate our workflow, but we feel that it is necessary to understand what we are showing in the phenotype UMAP (Fig. 4C). Furthermore, showing it only in the supplemental material would not save much space in the main figure due to the figure panel geometry.

-Figure 5B, the legend indicates the z-score from dark blue to dark red, but in the figure only light red are shown. I understand the authors showed selected library elements, but are there any knockdowns that consistently increased overall ISG expression?

Response: The reviewer is correct that no factor vastly increased overall ISG expression in bystander activated cells when knocked down. For this plot, we chose a symmetrical color scale in which neutral white represents no change in ISG expression, so there are indeed no host targets that would be showing “dark red” ISG levels. See also panel A of this figure for a sense of the asymmetry of the distribution, with only left-sided outliers.

-One of the main findings in this paper is the dependency on NFKBIA by SARS-CoV-2. The authors and Simoneau et al. demonstrated such dependency is likely the result of disruption on the NFkB nuclear translocation, highlighting a nonlinear feedback loop by NFKBIA. Besides NFKBIA, TNFAIP3 (A20) is also well-known inhibitor of NFkB activation, was it one of the Perturb-seq library targets? If so, does TNFAIP3 knockdown also results in similar phenotype?

Response: We thank the reviewer for this question and agree that it would be interesting to know if TNFAIP3 knockdown phenocopied NFKBIA knockdown’s effect on SARS-CoV-2 infection. Unfortunately, TNFAIP3 was not a target in our Perturb-seq library so we cannot make a statement to that effect.

Reviewer #3

The authors have sufficiently addressed all concerns.

Response: We thank the reviewer for their positive assessment of our manuscript!